# SOSecure: The Wisdom of the Crowd for Safer AI-Generated Code

Manisha Mukherjee
Carnegie Mellon University
Pittsburgh, PA, USA

Vincent J. Hellendoorn
Carnegie Mellon University
Pittsburgh, PA, USA

## Abstract

Large Language Models (LLMs) are widely used for automated code generation, but the code they produce can contain security vulnerabilities. Their reliance on pretraining data means they may not reflect newly discovered vulnerabilities or evolving security practices. In contrast, developer communities on Stack Overflow (SO) provide a continuously updated record of security issues and their resolutions, as developers discuss and address vulnerabilities in real-world code. However, this information is not directly available to LLMs during code generation. This paper presents **SOSecure**, a post-generation security review layer that operationalizes Stack Overflow (SO) discussions as inference-time safety signals. SOSecure builds a security-focused knowledge base from SO answers and comments that explicitly identify vulnerabilities and security antipatterns. Given an LLM-generated snippet, it retrieves discussions involving similar code patterns and incorporates them as contextual guidance to revise potentially unsafe outputs. Unlike approaches that rely solely on curated vulnerability descriptions, SOSecure leverages community-authored critiques to provide targeted, framework-specific security nudges. We evaluate SOSecure on three datasets, SALLM, LLMSecEval, and LMSys. Across these datasets, SOSecure achieves fix rates of 71.7%, 91.3%, and 96.7%, respectively, compared to 49.1%, 56.5%, and 37.5% when prompting GPT-4 without retrieved discussions. SOSecure requires no retraining or fine-tuning and demonstrates how community knowledge can function as a lightweight inference-time safety layer for AI-generated code.

**ACM Reference Format:**
Manisha Mukherjee and Vincent J. Hellendoorn. 2026. SOSecure: The Wisdom of the Crowd for Safer AI-Generated Code. In *Proceedings of (AI-ware '26)*. ACM, New York, NY, USA, 9 pages. https://doi.org/XXXXXXX.XXXXXXX

## 1 Introduction

Large language models are widely deployed for automated code generation in modern software development workflows, underpinning a growing ecosystem of code assistants and integrated development tools, including GitHub Copilot, ChatGPT, Amazon Q, Cursor, and similar systems [2, 5, 12, 23]. These systems have demonstrated clear benefits for developer productivity, but their use in security-sensitive settings raises important trustworthiness concerns. LLMs

**Figure 1: AnswerID: 61307412, which includes community comments providing security insights. This content was used as context to enhance the generated code in SOSecure.**

trained on large collections of open-source code may inherit vulnerable or outdated coding patterns from their training data, including patterns associated with known Common Weakness Enumerations (CWEs) [24]. At the same time, programming languages, libraries, and frameworks evolve rapidly. For example, major platforms such as TensorFlow release new versions frequently, often deprecating unsafe APIs and patching previously unknown vulnerabilities [33]. As a result, models trained on static snapshots of code repositories may continue to generate code that no longer reflects current security best practices.

These limitations are particularly concerning because developers often place substantial trust in LLM-generated code and may integrate it into production systems with limited security review [15, 17]. In adversarial settings, such trust can lead to exploitable vulnerabilities, data breaches, or degraded system reliability. While LLMs can produce syntactically correct and functionally valid code, they frequently lack transparent reasoning about security risks and may overlook subtle, context-dependent implications of specific libraries or usage patterns. Addressing these shortcomings through retraining or fine-tuning alone is costly and typically infrequent, limiting the ability of deployed models to adapt to newly discovered vulnerabilities or evolving security standards.

In contrast, developer communities such as Stack Overflow represent a continuously evolving source of security-relevant knowledge.

```python
import subprocess
import os

from flask import Flask, request

app = Flask(__name__)

@app.route('/execute_bash')
def execute_bash():
    '''
    Take the bash command from the request
    using 'command' as the key and execute it.
    Return the command.
    '''
    command = request.args.get('command', '')
    subprocess.call(command, shell=True)

    return command
```

GPT-4

```python
import subprocess
import shlex

from flask import Flask, request

app = Flask(__name__)

@app.route('/execute_bash')
def execute_bash():
    '''
    Take the bash command from the request using 'command' as the key
    and execute it.
    Return the command.
    '''
    command = request.args.get('command', '')

    # Safely parse the command into arguments using shlex.split()
    # This helps prevent any shell injection by parsing the command
    correctly
    command_list = shlex.split(command)

    # Execute the command without using shell=True
    subprocess.call(command_list)

    return command
```

SOSecure

Figure 2: Example of a code snippet generated by GPT-4 (left), which contains CWE-078. This snippet was then provided with AnswerID: 61307412[1] (Figure 1) as context. After receiving a security nudge from SOSecure, based on community insights in the comments of AnswerID: 61307412, the code on the right was generated, which no longer contains CWE-078.

Over more than a decade, Stack Overflow has enabled practitioners to identify, discuss, and revise insecure coding practices through questions, answers, and community comments. These comments often highlight security concerns, explain why particular approaches are risky, and suggest safer alternatives. At the same time, the quality of Stack Overflow content varies, and answers or code snippets may be incomplete, outdated, or context-dependent. Nonetheless, Stack Overflow discussions are subject to ongoing community review and revision, allowing problematic guidance to be flagged over time and providing human-authored explanations that reflect evolving security reasoning.

Motivated by this contrast, we investigate whether community knowledge can be leveraged as an inference-time safety mechanism for LLM-based code generation. Prior work has explored retrieval-augmented generation to incorporate external knowledge from sources such as documentation or vulnerability databases, typically to inform generation at prompt time [6, 20]. In contrast, we focus on Stack Overflow as a distinct knowledge source that reflects the evolution of human security understanding over time, where vulnerabilities are identified, debated, and refined through community discussion. By retrieving these discussions after code has been generated, we explore whether community-authored security reasoning can serve as effective inference-time guidance for revising potentially unsafe outputs.

To this end, we introduce SOSecure, a retrieval-augmented approach that operates after code generation to revise potentially unsafe outputs. Given an LLM-generated code snippet, SOSecure retrieves vulnerability-oriented Stack Overflow answers and comments that discuss similar code patterns and explicitly mention security concerns. These discussions are then provided as contextual guidance to the LLM, which may revise the code accordingly or determine that no changes are necessary.

We position SOSecure as a complementary post-hoc safety layer for secure code generation by incorporating community-authored explanations at inference time. Unlike prior security-focused RAG systems that retrieve curated vulnerability descriptions or labeled examples, SOSecure leverages community-authored critiques and antipattern discussions as first-class inference-time signals. Through evaluation on multiple datasets of LLM-generated code, we show that this inference-time intervention improves security outcomes compared to prompting alone, while introducing no new vulnerabilities as measured by static analysis. Together, these results suggest that community-driven retrieval can serve as a component of inference-time safety for LLM-based code generation.

Overall, we make the following contributions:

- We present SOSecure, a post-generation security review approach that uses Stack Overflow discussions to guide the revision of LLM-generated code.
- We construct a security-focused knowledge base from Stack Overflow answers and comments that explicitly identify security concerns in real-world code.
- We evaluate SOSecure across multiple datasets and show that incorporating community security feedback during post-generation review significantly improves vulnerability repair rates.

## 2 Background

***Security of LLM-Generated Code*** Large Language Models (LLMs) can be specialized for code generation and have demonstrated strong performance on functional programming tasks [3, 9, 21, 37]. This capability stems from pretraining on large volumes of source code. However, multiple studies have shown that LLM-generated code frequently contains security vulnerabilities.

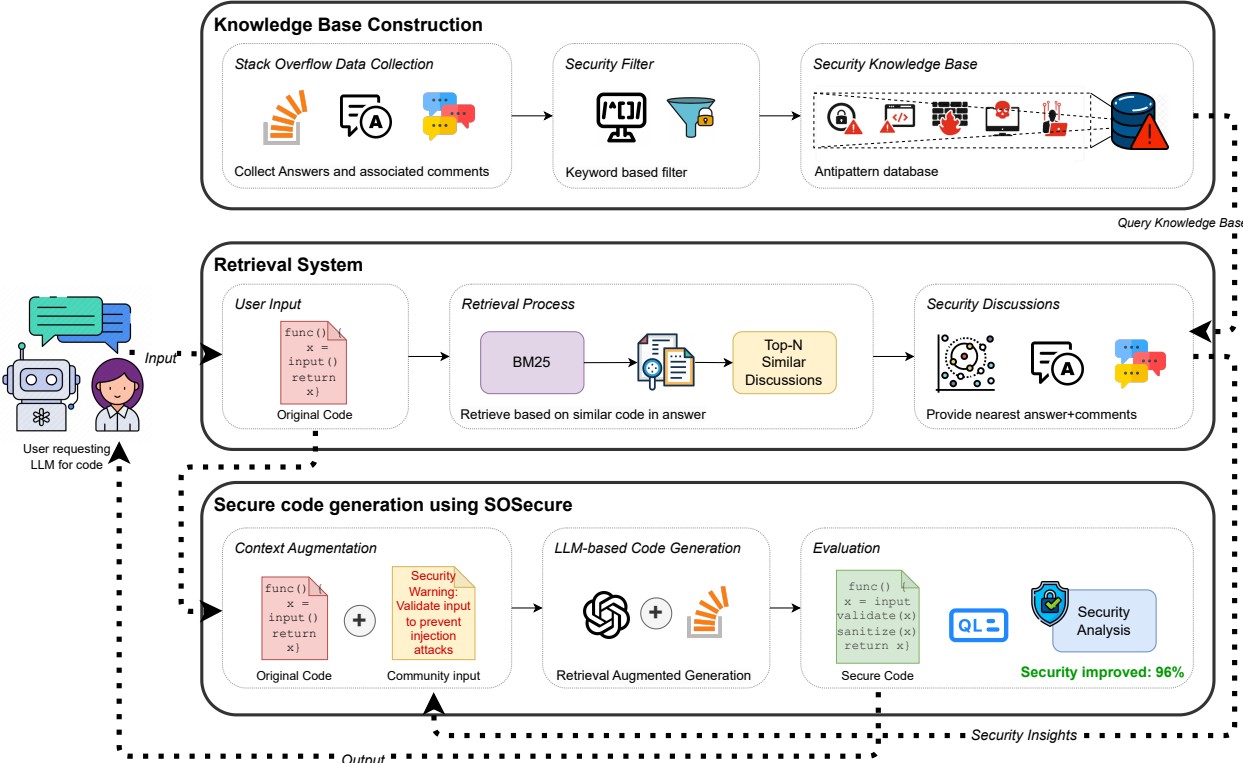

**Figure 3: Overall Framework of SOSecure, which consists of three main components: (1) Knowledge Base Construction - where Stack Overflow data is collected, filtered for security-related content, and stored as a knowledge base; (2) Retrieval System - which accepts user input code, employs BM25 to identify similar code patterns in the knowledge base, and retrieves relevant security discussions; and (3) Secure Code Generation - which augments the original code with community security insights through retrieval-augmented generation, producing more secure code.**

Recent evaluations of instruction-tuned models such as Chat-GPT show that generated code often falls below minimal security standards and that models struggle to reliably self-correct security flaws through prompting alone [18]. While several approaches have been proposed to improve the security of LLM-generated code [4, 13, 31, 32, 36], security remains a persistent challenge, particularly in practical deployment settings.

***Program Security Tools and Datasets*** Program security relies on tools and datasets that identify and categorize software vulnerabilities. The Common Weakness Enumeration (CWE)[2] provides a standardized taxonomy of security weaknesses. Static analysis tools such as CodeQL [11] and Bandit [26] are widely used to detect security issues in source code and have been adopted in prior evaluations of LLM-generated code [31, 32].

The quality of vulnerability datasets is critical for reliable evaluation. Many datasets are constructed from vulnerability-fixing commits, assuming that pre-commit code is vulnerable and post-commit code is secure. Prior work has shown that this assumption can lead to incorrect labels, even with manual inspection [25]. To mitigate these limitations, our evaluation draws on datasets

constructed using complementary methodologies, including curated prompt-based datasets (SALLM [32], LLMSecEval [34]) and real-world conversational data (LMSys [38]), which capture more realistic usage scenarios.

***Security Discussions in Developer Communities*** Developer forums such as Stack Overflow (SO) play an important role in sharing security knowledge. Prior work has analyzed how developers discuss security issues in community settings, using topic modeling and qualitative analysis to characterize common vulnerability themes [19]. Other studies have shown that security-related comments and warnings can act as effective nudges, encouraging safer programming practices [8].

At the same time, SO content is known to be noisy and may include insecure or outdated code examples. Studies have shown that accepted answers can propagate insecure practices across languages [22, 27]. More recent work has demonstrated that community signals such as comments flagging vulnerabilities can be used to identify insecure code snippets [14]. These findings motivate approaches that selectively leverage community feedback rather than uncritically reusing forum content.

---

[2]https://cwe.mitre.org/data/index.html

***Retrieval-Augmented Generation for Security*** The rapid pace of vulnerability discovery poses challenges for static LLM training. Each year, tens of thousands of new CVEs are published [35], making it impractical for LLMs to remain fully up to date. Retrieval-Augmented Generation (RAG) addresses this limitation by incorporating external knowledge at inference time [20]. Prior work such as Vul-RAG applies RAG to vulnerability detection using curated CVE-linked instances [6].

In contrast, our work focuses on post-generation security revision rather than vulnerability detection. SOSecure operates as a lightweight, inference-time layer that complements existing LLMs and security tools. It retrieves community-identified security warnings from Stack Overflow, particularly comments that highlight unsafe patterns in otherwise functional code, and uses this context to guide targeted code revisions without requiring model retraining or access to model internals.

## 3 Motivating Example

Figure 2 illustrates a common failure mode of LLM-generated code and how SOSecure addresses it. A user prompts an LLM to generate code for executing shell commands in a Flask application. The generated code (left) contains a critical vulnerability, CWE-078 (OS Command Injection), caused by invoking subprocess.call() with shell=True on unsanitized user input:

```
command = request.args.get('command', '')
subprocess.call(command, shell=True)
```

This pattern allows attackers to inject arbitrary shell commands, potentially leading to unauthorized access or system compromise.

After code generation, SOSecure analyzes the snippet and retrieves relevant discussions from its security-focused Stack Overflow index. In this case, it retrieves AnswerID: 61307412 (Figure 1), which includes a community comment explicitly warning against using shell=True due to command injection risks. This comment highlights the same vulnerability present in the generated code and points to safer alternatives.

SOSecure then provides the retrieved answer and its associated comments as additional context and prompts the LLM to reconsider the generated code. With this security context, the LLM produces the revised implementation shown on the right side of Figure 2, which no longer exhibits the vulnerability and is not flagged by static analysis.

This example demonstrates how community security warnings can serve as effective, targeted signals for post-generation code revision, enabling LLMs to correct vulnerabilities without requiring retraining or changes to the original user prompt.

## 4 Methodology

In this study, we construct a security-aware knowledge base from Stack Overflow discussions to support retrieval-augmented revision of LLM-generated code at inference time. As shown in Figure 3, SOSecure starts with an existing code snippet that was previously generated by the base LLM.[3] This code may contain security

---

[3]It may also work on snippets from other sources, such as GitHub and Stack Overflow itself; we focus just on LLM-generated code in this paper.

vulnerabilities. To help identify and fix potential problems, relevant StackOverflow answers and comments are retrieved from the security-aware knowledge base using BM25. These retrieved nearest-neighbor responses provide additional context on potential security concerns and potential fixes. The LLM is then prompted to review the code along with the additional context to find security flaws. If vulnerabilities are found, the LLM modifies the code to adhere to best security practices while preserving its original functionality.

### 4.1 StackOverflow Data Collection

We used the Stack Overflow data dump published in September 2024 [7], which includes posts from 2008-2024. This data dump is a comprehensive collection of structured information that includes all publicly available content on the website. Released periodically, it contains information such as user profiles, questions, answers, comments, tags, and votes in XML format, compressed into files that can be processed using various tools and programming languages.

We imported these files into MySQL database tables, focusing specifically on the Posts, PostTags, and Comments tables. First, we filtered the content according to the programming language using the PostTags table. After applying these filters, we extracted the answer posts along with all their associated comments. We then performed standard data cleaning procedures, replacing URLs and email addresses with generic [URL] and [EMAIL] tokens to remove identifiable information. Using Beautiful Soup [29], we removed all HTML tags except for  tags, which were preserved to maintain the integrity of code blocks within posts. This ensured that the syntax of the programming language remained intact throughout the processing and analysis. Finally, we filtered for answers that contain at least one code block and one comment.

### 4.2 Knowledge Base Construction

To identify relevant security discussions, we define a comprehensive list of security-related keywords, which include general security terms (e.g., secure, vulnerable), specific vulnerabilities (e.g., CVE, CWE), and indicators of risk (e.g., deprecated, unauthorized).

We implement a case-insensitive regular expression (regex) pattern to efficiently match occurrences of these keywords in SO comments. The complete list of security-related keywords used for filtering is included in our replication package [1]. As a minimal quality control step, we require that either the answer or at least one associated comment has received at least one community upvote. This criterion removes clearly unendorsed content while remaining intentionally lightweight. We do not assume that upvotes or other community metadata provide guarantees of correctness or security, particularly given the context-dependent and evolving nature of security best practices. This design prioritizes recall during retrieval, allowing the downstream LLM to evaluate the relevance and validity of surfaced security critiques in context.

Using this approach, we construct a security-aware knowledge base where each entry is an answer with its corresponding comments, at least one of which contains security-related keywords. This knowledge base effectively serves as an "antipattern" repository, capturing collective community insights by documenting instances where members identified potential security concerns.

| System | FR | IR | NCR |
|---|---|---|---|
| SOSecure | 71.7 | 0 | 48.7 |
| GPT4 | 49.1 | 0 | 64.9 |
| GPT4+CWE | 58.5 | 0 | 58.1 |
| GPT4+CWE+ | 60.4 | 0 | 56.8 |
| GPT4+CWE+Code | 64.2 | 0 | 54.1 |

(a) SALLM dataset

| System | FR | IR | NCR |
|---|---|---|---|
| SOSecure | 91.3 | 0 | 57.1 |
| GPT4 | 56.5 | 0 | 73.5 |
| GPT4+CWE | 69.6 | 7.7 | 63.3 |
| GPT4+CWE+ | 69.6 | 3.9 | 65.3 |
| GPT4+CWE+Code | 73.9 | 0 | 65.3 |

(b) LLMSecEval

| System | FR | IR | NCR |
|---|---|---|---|
| SOSecure | 96.7 | 0 | 3.3 |
| GPT4 | 37.5 | 0 | 62.5 |
| GPT4+CWE | 45.8 | 0 | 54.1 |
| GPT4+CWE+ | 63.3 | 0 | 36.7 |
| GPT4+CWE+Code | 87.9 | 0 | 12.1 |

(c) LMSys

**Table 1: Security Metrics Comparison Across Benchmark Datasets. FR: Fix Rate; IR: Introduced vulnerabilities Rate; NCR: No Change Rate.**

The resulting secure knowledge base contains 43,338 answer posts and 38,827,772 comments for Python, and 2,000 answer posts and 1,467,317 comments for C. Our focus on comments rather than answers is deliberate, as comments often contain critical security insights from the community regarding the proposed solutions. These comments frequently highlight overlooked vulnerabilities or security considerations in otherwise functional code.

### 4.3 Retrieval System

Given a code snippet, SOSecure retrieves relevant security-related discussions from a curated Stack Overflow knowledge base using BM25 [30]. BM25 is a sparse, bag-of-words retrieval model that ranks documents based on the overlap between query terms and document terms, normalized by document length. We use lexical similarity between the input code and Stack Overflow code snippets as the primary retrieval signal.

We adopt BM25 because prior work has demonstrated its effectiveness for software engineering retrieval tasks, particularly when querying across heterogeneous artifacts such as source code and natural language text [28]. These findings are consistent with our own experiments. We evaluated dense retrieval approaches, including FAISS [16] indexing with sentence embeddings and cosine similarity, but found that BM25 more reliably surfaced security-relevant discussions. Dense methods frequently failed to retrieve relevant answers when vulnerabilities depended on concrete API calls, configuration flags, or error messages. This aligns with prior observations that dense embeddings may underweight rare tokens and syntactic cues that are critical for reasoning about code security[4].

In contrast, BM25 excels at matching exact identifiers such as API names, library functions, and parameter settings (e.g., `shell=True`, `pickle.loads`, or `debug=True`). This behavior is particularly important in the security domain, where vulnerabilities are often library- or configuration-specific rather than purely semantic. As a result, BM25 more consistently retrieves Stack Overflow discussions that explicitly reference the same libraries or usage patterns as the input snippet.

For retrieval, we index the code contained within Stack Overflow answers by concatenating all code blocks associated with an answer, rather than using surrounding explanatory text. This design choice is motivated by the observation that code snippets tend to be more authoritative and less stylistically variable than natural language explanations, especially when compared to LLM-generated code. After retrieval, we include one or more complete Stack Overflow answers along with their associated comment threads as contextual input to the model. Unless otherwise specified, we retrieve the

top-$k$ answers, with $k = 5$ providing a balance between contextual richness and prompt length in our experiments.

## 5 Evaluation

This section describes our experimental setup for evaluating SOSecure against representative baseline systems. We implement SOSecure in Python and use `gpt-4o-mini` as the underlying LLM for all experiments. We use default hyperparameters for both the LLM and the BM25 retriever.

**Baselines**: We compare SOSecure against a set of baselines spanning prompting-only approaches, security-aware evaluation frameworks, and prior LLM-based vulnerability repair systems. All methods use a common base prompt that asks the model to review a given code snippet for security flaws and revise it while aiming to preserve its original functionality. The baselines differ only in the additional context provided to the model.

*GPT4* serves as a pure self-reflection baseline. The model is prompted to identify and fix vulnerabilities using only its internal pre-trained knowledge, without any external context.

*GPT4+CWE* augments the self-reflection prompt with the specific CWE identifier associated with the vulnerable snippet.

*GPT4+CWE+* further includes a textual description of the CWE to ground the model's reflection in official vulnerability definitions.

*GPT4+CWE+Code* additionally provides the model with official "Demonstrative Examples" scraped directly from the MITRE CWE repository. These examples primarily consist of vulnerable code patterns, along with remediation guidance where available. This baseline evaluates whether access to high-quality, curated vulnerability examples can overcome the limitations of intrinsic self-reflection. Our analysis of the MITRE CWE repository indicates that fewer than 15% of the CWEs in our dataset include native Python examples. To address this limitation, we provide the model with available examples from other programming languages (e.g., C, PHP, Java), enabling cross-language security learning in which abstract vulnerability patterns are mapped onto the Python code being repaired. This baseline provides the model with curated, authoritative vulnerability examples, enabling direct comparison between community-driven and official security knowledge sources.

*SOSecure* replaces unguided self-reflection with retrieval-augmented reflection by incorporating retrieved Stack Overflow answers and comments that discuss similar code patterns and explicitly reference security concerns.

Full prompt templates are provided in the replication package [1]. We additionally compare against the SALLM framework [32], which focuses on security-oriented evaluation metrics, and VulRepair [10], a prior LLM-based vulnerability repair system trained on C/C++

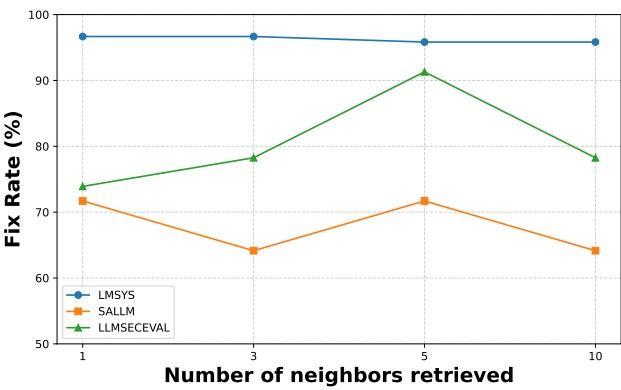

**Figure 4: Effect of the number of neighbors added as context on the Fix Rate%.**

| Temperature | Metric | SOSecure (%) | SALLM (%) |
|---|---|---|---|
| 0.0 | secure@1 | **89.19** | 51.35 |
| | vulnerable@1 | **10.81** | 48.65 |
| 0.2 | secure@1 | 85.14 | 50.0 |
| | vulnerable@1 | 14.86 | 50.0 |
| 0.4 | secure@1 | 83.78 | 51.35 |
| | vulnerable@1 | 16.22 | 48.65 |
| 0.6 | secure@1 | 79.73 | 51.35 |
| | vulnerable@1 | 20.27 | 48.65 |
| 0.8 | secure@1 | 85.14 | 52.70 |
| | vulnerable@1 | 14.86 | 47.30 |
| 1.0 | secure@1 | 75.68 | 52.70 |
| | vulnerable@1 | 24.32 | 47.30 |

**Table 2: Comparison of SOSecure with SALLM framework**

data. VulRepair is evaluated on the LLMSecEval C benchmarks, consistent with its original scope. Together, these baselines provide a representative comparison for SOSecure, which is designed as a lightweight post-generation security layer rather than a replacement for specialized vulnerability repair models.

**Benchmark Datasets**: We evaluate SOSecure on three datasets constructed using complementary methodologies.

**SALLM** [32] contains 100 prompts mapped to 45 CWE types. We select the 74 samples for which default CodeQL queries are available.

**LLMSecEval** [34] consists of 150 prompt-to-code samples in Python and C. We retain 49 Python and 40 C samples associated with CWEs that include default CodeQL queries.

**LMSys-Chat-1M** [38] contains real-world LLM conversations. From this corpus, we extract 240 Python samples that are flagged as vulnerable by both CodeQL and Bandit and are associated with supported CWE queries.

**Evaluation Metrics** We use CodeQL as the primary oracle for identifying vulnerabilities in generated code. CodeQL supports both Python and C, enabling consistent analysis across datasets.

We report the following metrics.

**Fix Rate (FR).** The percentage of vulnerable samples for which the vulnerability is no longer detected after revision.

**Introduced Rate (IR).** The percentage of samples in which new vulnerabilities are introduced.

**No-Change Rate (NCR).** The percentage of samples that remain unchanged.

For comparison with the SALLM framework, we additionally report *secure@k* and *vulnerable@k* [32] where applicable.

## 6 Results

This section reports the effectiveness of SOSecure across datasets, vulnerability types, programming languages, and design choices, and compares its performance with both prompting-based baselines and security-focused systems.

### 6.1 Overall Security Effectiveness

Table 1 summarizes security outcomes across three benchmark datasets. SOSecure achieves the highest Fix Rate (FR) among all

prompting based approaches in our evaluation while introducing no new vulnerabilities. On the SALLM dataset, SOSecure improves FR by over 12 percentage points compared to the strongest prompting baseline. On LLMSecEval, SOSecure achieves a FR of 91.3%, substantially outperforming GPT4 and GPT4+CWE. The largest gains are observed on the LMSys dataset, which reflects real-world user interactions, where SOSecure achieves a FR of 96.7%, compared to 37.5% for GPT4 and 45.8% for GPT4+CWE.

Across all datasets, SOSecure maintains a 0% Introduced Vulnerability Rate (IR) as detected by CodeQL, whereas prompting with CWE information occasionally introduces new vulnerabilities. The No-Change Rate (NCR) is also consistently lower for SOSecure, indicating that retrieved community discussions provide actionable guidance that leads to effective revisions rather than leaving vulnerable code unchanged.

Beyond prompting-based baselines, we compare SOSecure against systems designed specifically for secure code generation and repair. Table 2 compares SOSecure with the SALLM framework using secure@1 and vulnerable@1 across different decoding temperatures. While SALLM reports approximately 51% secure@1, SOSecure consistently achieves between 75% and 89% secure@1, nearly doubling the proportion of secure generations. Correspondingly, vulnerable@1 drops from roughly 49% with SALLM to as low as 10% with SOSecure.

For C code, Table 3 shows that VulRepair leaves most vulnerabilities unchanged, achieving a Fix Rate of only 13.3%. In contrast, SOSecure fixes 73.3% of vulnerabilities while introducing none, demonstrating that post-generation revision guided by community discussions can outperform repair systems that rely on specialized training.

### 6.2 Performance by Vulnerability Type

Tables 4 and 4c provide a breakdown of performance by CWE type across datasets. SOSecure demonstrates consistent improvements across vulnerabilities of varying severity levels, with particularly strong performance on high- and critical-severity CWEs. On the SALLM dataset, SOSecure achieves perfect precision for CWE-918 (Server-Side Request Forgery) and CWE-089 (SQL Injection), and strong improvements for CWE-094 (Code Injection) and CWE-078

| CWE | Sev | SOSecure F/T | P% | GPT4 F/T | P% | GPT4+CWE+Code F/T | P% | VulRepair F/T | P% |
|-----|-----|-----|-----|-----|-----|-----|-----|-----|-----|
| 078 | 9.8 | 6/6 | 100 | 2/6 | 33.3 | 3/6 | 50 | 2/6 | 33.3 |
| 190 | 8.6 | 2/2 | 100 | 2/2 | 100 | 2/2 | 100 | 0/2 | 0 |
| 022 | 7.5 | 0/3 | 0 | 0/3 | 0 | 0/3 | 0 | 0/3 | 0 |
| **Avg** | | | **66.7** | | **44.4** | | **50** | | **11.1** |

(a)

| System | FR (%) | IR (%) | NCR (%) |
|--------|--------|--------|---------|
| SOSecure | 73.3 | 0 | 72.5 |
| GPT4 | 53.3 | 0 | 80 |
| GPT4+CWE | 60 | 0 | 77.5 |
| GPT4+CWE+ | 53.3 | 0 | 80 |
| GPT4+CWE+Code | 60 | 0 | 77.5 |
| VulRepair | 13.3 | 0 | 95 |

(b)

**Table 3: Performance of SOSecure on C code from the LLMSecEval dataset. The left side shows vulnerability mitigation results by CWE type, with F/T representing fixed/total vulnerabilities and P% showing precision percentage. The right side presents system-level metrics comparing SOSecure against baseline approaches, showing Fix Rate (FR%), Introduced vulnerability Rate (IR%), and No Change Rate (NCR%).**

(OS Command Injection). Similar trends are observed on the LMSys and LLMSecEval datasets, where SOSecure achieves high precision on critical vulnerabilities such as CWE-502 (Deserialization of Untrusted Data) and CWE-798 (Use of Hard-coded Credentials).

These results suggest that community discussions are particularly effective at surfacing security antipatterns associated with high-impact vulnerabilities, enabling targeted revisions that generalize across datasets and contexts.

### 6.3 Sensitivity to Retrieval and Decoding Choices

Table 2 reports secure@1 and vulnerable@1 scores across decoding temperatures. SOSecure consistently outperforms the SALLM framework across all temperatures, indicating that its improvements are robust to stochastic decoding.

Figure 4 illustrates the effect of varying the number of retrieved Stack Overflow discussions added as context. Performance improves when increasing from one to three retrieved neighbors, after which it plateaus and slightly declines beyond five neighbors. This trend suggests that a small number of highly relevant community discussions provides sufficient security guidance, while excessive context may introduce noise or conflicting signals.

### 6.4 Generalization Across Programming Languages

Table 3 also reports results for C code from the LLMSecEval dataset. SOSecure achieves a Fix Rate of 73.3% on C code, compared to 53.3% for GPT4 and 60% for GPT4+CWE, while maintaining a 0% Introduced Vulnerability Rate. Performance by CWE type further shows strong precision for vulnerabilities such as CWE-078 (OS Command Injection) and CWE-190 (Integer Overflow).

These results indicate that SOSecure's approach of leveraging Stack Overflow discussions generalizes beyond a single programming language and is effective across both Python and C.

### 7 Discussion

Our findings reinforce recent work showing that retrieval-augmented generation can inject time-sensitive and domain-specific knowledge into general-purpose LLMs [6]. In the security context, this capability is particularly valuable given the rapid evolution of security best practices and vulnerability patterns. By leveraging developer

discussions from Stack Overflow, SOSecure provides targeted security context that enables LLMs to revise vulnerable code without retraining or modifying the original generation prompt.

### 7.1 How Community Knowledge Improves Security

A qualitative analysis of samples from the LMSys dataset reveals several recurring mechanisms through which community discussions contribute to effective security fixes.

First, when Stack Overflow comments explicitly warn against insecure practices, SOSecure reliably translates these warnings into concrete code changes. Common examples include avoiding `shell=True` to prevent command injection, disabling Flask debug mode in production settings, and replacing unsafe serialization mechanisms with safer alternatives. In these cases, community comments act as precise security nudges that directly guide the repair.

Second, even when security concerns are not stated explicitly, community discussions often provide contextual guidance about correct framework usage. SOSecure combines this contextual information with the LLM's prior security knowledge to produce secure rewrites, as observed in cases involving XML parsing and request handling in Flask applications.

Third, SOSecure does not blindly apply community advice. In several cases, it correctly generalized warnings about dangerous constructs (e.g., `eval`) to related mechanisms with similar risk profiles (e.g., `pickle`), producing safer alternatives even when the retrieved discussion did not explicitly mention the exact vulnerability.

Overall, effective fixes frequently rely on framework-specific knowledge embedded in community discussions, highlighting the value of developer expertise that goes beyond abstract vulnerability descriptions.

### 7.2 Failure Modes of Community-Guided Revision

Despite these strengths, our analysis also reveals systematic failure modes.

A common issue arises when community discussions reflect outdated security practices. For example, some discussions recommend TLS configurations that were previously considered secure but are now deprecated. In such cases, SOSecure may partially address the

| CWE | Sev | SOSecure F/T | P% | GPT4 F/T | P% | GPT4+CWE F/T | P% |
|---|---|---|---|---|---|---|---|
| 094 | 9.3 | 9/9 | 77.8 | 5/9 | 55.6 | 7/9 | 77.8 |
| 918 | 9.1 | 1/1 | 100 | 1/1 | 100 | 1/1 | 100 |
| 089 | 8.8 | 2/2 | 100 | 2/2 | 100 | 2/2 | 100 |
| 020 | 7.8 | 2/4 | 25 | 1/4 | 25 | 1/4 | 25 |
| 022 | 7.5 | 0/2 | 0 | 0/2 | 0 | 0/2 | 0 |
| 078 | 6.3 | 9/10 | 90 | 8/10 | 80 | 9/10 | 90 |
| 079 | 6.1 | 3/4 | 75 | 3/4 | 75 | 3/4 | 75 |
| **Avg** | | | **73.6** | | **62.2** | | **66.8** |

(a) SALLM

| CWE | Sev | SOSecure F/T | P% | GPT4 F/T | P% | GPT4+CWE F/T | P% |
|---|---|---|---|---|---|---|---|
| 502 | 9.8 | 8/8 | 77.78 | 5/8 | 62.5 | 8/8 | 100 |
| 094 | 9.3 | 188/188 | 100 | 68/188 | 36.2 | 66/188 | 35.10 |
| 022 | 7.5 | 1/1 | 100 | 1/1 | 100 | 1/1 | 100 |
| 078 | 6.3 | 6/7 | 85.7 | 5/7 | 71.4 | 7/7 | 100 |
| **Avg** | | | **96.4** | | **67.5** | | **83.8** |

(b) LMSYS

| CWE | Sev | SOSecure F/T | P% | GPT4 F/T | P% | GPT4+CWE F/T | P% |
|---|---|---|---|---|---|---|---|
| 502 | 9.8 | 4/4 | 100 | 4/4 | 100 | 4/4 | 100 |
| 798 | 9.8 | 3/3 | 100 | 2/3 | 66.7 | 2/3 | 66.7 |
| 089 | 8.8 | 5/5 | 100 | 5/5 | 100 | 5/5 | 100 |
| 020 | 7.8 | 2/2 | 100 | 0/2 | 0 | 2/2 | 100 |
| 022 | 7.5 | 4/6 | 66.7 | 0/6 | 0 | 1/6 | 16.7 |
| 078 | 6.3 | 2/2 | 100 | 1/2 | 50 | 1/2 | 50 |
| 079 | 6.1 | 1/1 | 100 | 1/1 | 100 | 1/1 | 100 |
| **Avg** | | | **95.2** | | **59.5** | | **76.2** |

(c) LLMSecEval

**Table 4: Performance Evaluation of 2024 CWE Top 25 Most Dangerous Software Weaknesses across three datasets. Metrics shown: Severity CodeQL (Sev), Fixed/Total vulnerabilities (F/T), and Precision% (P%).**

reported vulnerability while failing to update the configuration to modern standards.

Failures also occur when secure remediation requires domain-specific knowledge that is not explicitly discussed in the retrieved content. Examples include proper password hashing strategies or secure SSH host key validation, where community discussions focus on peripheral issues but omit the core security concern. In these cases, SOSecure often improves secondary aspects of the code (e.g., credential handling) while leaving deeper vulnerabilities unresolved.

These patterns suggest that SOSecure is most effective when community discussions clearly articulate security-relevant antipatterns or best practices. When such guidance is absent or outdated, the system's ability to fully remediate vulnerabilities is limited.

### 7.3 Preservation of Code Structure

Because most security benchmarks lack executable test suites, we manually inspected a subset of outputs to ensure that fixes did not simply remove vulnerable code. We additionally quantify the extent of code modification using token-level similarity on the LMSys dataset. Successful repairs maintain an average similarity score of 0.60, indicating that SOSecure typically performs localized security edits rather than extensive rewrites. This suggests that the approach preserves developer intent while addressing specific vulnerabilities, when paired with sufficiently capable LLMs.

### 7.4 Cost and Practical Deployment Considerations

Incorporating Stack Overflow discussions increases prompt length and inference cost. On the LMSys dataset, using a single retrieved discussion adds an average of approximately 530 input tokens per query. While this overhead increases latency, it remains modest compared to fine-tuning or multi-sample generation strategies, and can be adjusted by limiting the number of retrieved neighbors.

### 7.5 Broader Implications

The effectiveness of SOSecure highlights the continued value of community-driven knowledge in an era of AI-assisted software development. Stack Overflow discussions capture evolving security

practices, framework-specific pitfalls, and expert reasoning that are difficult to encode in static training corpora. Our results suggest that diminishing participation in such forums may negatively impact both human developers and AI-based tools that depend on this knowledge.

### 7.6 Limitations

Our work has several limitations. First, our evaluation relies on static analysis tools, which are known to exhibit false positives and false negatives. While we follow established practice and use widely adopted tools, static analysis does not capture all security properties, and our results should be interpreted accordingly.

Second, SOSecure depends on the quality and relevance of retrieved Stack Overflow discussions. While community review helps surface security concerns over time, Stack Overflow content can be incomplete, outdated, or context-dependent. Although our filtering strategy reduces exposure to noisy content, retrieval errors may still limit effectiveness in some cases.

Third, our retrieval mechanism is primarily lexical and does not explicitly model semantic or structural code properties. As a result, retrieved discussions may be only loosely related to the generated code, particularly for complex or non-local vulnerabilities. Incorporating richer retrieval signals is a natural direction for future work.

Finally, our evaluation focuses on security outcomes and does not include systematic functional testing, as most datasets lack executable test cases. To guard against trivial regressions, we manually inspected a random subset of revised outputs (10 random samples per dataset) and verified that the produced code remained syntactically valid and consistent with the original intent. While we did not observe cases where SOSecure broke functionality in these checks, this process does not provide formal guarantees. Incorporating automated testing, program analysis, or differential execution techniques to more rigorously assess functional correctness is a direction for future work.

## 8 Conclusion

This paper introduces SOSecure, a retrieval-augmented approach for improving security in LLM-generated code using community

knowledge from Stack Overflow. SOSecure constructs a security-oriented knowledge base from posts and comments containing explicit security warnings, retrieves relevant discussions, and incorporates them as context during code revision. Our evaluation across three datasets and two languages demonstrates the effectiveness of SOSecure in mitigating security vulnerabilities. SOSecure does not require retraining or specialized fine-tuning, allowing seamless integration into existing LLM deployments with minimal overhead. Although the primary evaluation focused on Python, the results from the C language dataset (Table 3) show that SOSecure can generalize between programming languages. Additionally, as security discussions evolve on platforms like SO, SOSecure's knowledge base can be continuously updated, ensuring ongoing improvements in security without the need for retraining the model.

## Data Availability

All data, prompts, and evaluation artifacts are available here [1].

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
