# OpenReview forum: "SOSecure: The Wisdom of the Crowd for Safer AI-Generated Code"
_ACM.org/AIWare/2026/Conference — AIware 2026_

### Official Review · Reviewer_ZpV4 · 2026-03-07

**Rating:** 4
**Confidence:** 4

**Review:**

## Pros

* The paper addresses a relevant and practical problem, namely improving the security of LLM generated code.

* The approach is simple and easy to integrate because it operates at inference time without requiring model retraining.

* The idea of leveraging community discussions from Stack Overflow as security guidance is interesting and practically motivated.

* The evaluation includes multiple datasets and shows substantial improvements in Fix Rate compared with prompting based baselines.



## Cons

* The novelty of the approach is limited. The method mainly follows a standard retrieval augmented generation pipeline, with the primary difference being the use of Stack Overflow discussions as the knowledge source.

* The construction of the security knowledge base relies on heuristic keyword filtering and minimal quality constraints. The paper does not provide evidence about the reliability or correctness of the retrieved security discussions.

* The evaluation relies entirely on static analysis tools such as CodeQL and Bandit as vulnerability oracles. The claim that the approach introduces no new vulnerabilities depends completely on these tools.

* The evaluation does not include systematic functional testing of the revised code, and token similarity does not guarantee preservation of functionality.

---
## Quality and Soundness

The methodology is clearly described, and the system pipeline is easy to follow. However, the evaluation relies entirely on static analysis tools for vulnerability detection. Since static analysis tools may produce false positives and false negatives, the reliability of the reported Fix Rate and Introduced Rate is uncertain. In addition, the absence of systematic functional testing makes it unclear whether the generated fixes preserve program behavior.

This paper uses model of gpt-4 is kinda oudated and cannot reflect the latest model capability. It is recommended for the authors to try more advanced models, like gpt-5+ and claude opus or sonnet 4.5+ for more convincing evaluation.

## Clarity

The paper is generally readable, and the overall pipeline can be understood from the figures and system description. However, several methodological details could be explained more precisely. In particular, the process of constructing the security knowledge base and the justification of the keyword filtering strategy are only briefly described. It is also not entirely clear how retrieved Stack Overflow discussions are incorporated into the prompt and how the model decides whether and how to modify the code. Providing more detailed examples of prompts and retrieval outputs would improve transparency and reproducibility.

## Originality

The originality of the work is limited. The overall framework follows a standard retrieval-augmented generation pipeline where external knowledge is retrieved and appended to the prompt to guide code revision. The main difference from prior work lies in the choice of Stack Overflow discussions as the knowledge source rather than curated vulnerability descriptions. The paper does not clearly identify what methodological challenges arise from using community discussions or what new techniques are introduced to address the noise and inconsistency of such sources.

## Significance

Improving the security of LLM-generated code is an important problem, and leveraging community discussions as a dynamic knowledge source is an interesting direction. However, the current evaluation does not fully demonstrate the robustness or practical applicability of the approach. The results rely heavily on static analysis tools and do not include systematic functional validation of the revised code. In addition, the experiments are conducted with relatively outdated base models, leaving it unclear whether the proposed approach would provide similar benefits when applied to stronger modern code generation models.

**Summary:**

This paper proposes SOSecure, a post-generation security revision framework that leverages Stack Overflow discussions as inference-time safety signals for LLM-generated code. Instead of relying solely on self-reflection or curated CWE descriptions, SOSecure constructs a security-oriented knowledge base from Stack Overflow answers and associated comments containing security-related keywords. Given an LLM-generated snippet, the system retrieves lexically similar code examples using BM25 and augments the prompt with corresponding community discussions to guide code revision. The evaluation is conducted on three datasets, including SALLM, LLMSecEval, and LMSys, across Python and C. Results show substantial improvements in Fix Rate compared to prompting only baselines and curated CWE-based augmentation, while maintaining zero introduced vulnerabilities according to static analysis. The paper argues that community knowledge can function as a lightweight inference-time security layer without retraining.

---

> ### Author Response · Authors · 2026-03-21
>
> Thank you for your positive assessment and helpful feedback. We address your concerns below.
>
> **On Originality**
> While SOSecure follows a retrieval-augmented paradigm, its key distinction is post-generation, code-conditioned retrieval. This allows the system to target vulnerabilities that arise in the generated code itself, rather than relying on the input prompt or predefined vulnerability categories. In addition, SOSecure retrieves community-authored critiques tied to concrete code patterns, rather than curated descriptions or examples. The contribution is therefore not simply the use of Stack Overflow as a data source, but the introduction of a post-generation, code-conditioned retrieval paradigm grounded in code-specific, practitioner-authored signals.
>
> **On Knowledge Base Construction**
> Our goal is to show that even lightweight filtering of community discussions can yield strong gains. While Stack Overflow may contain noise, the empirical results indicate that the retrieved signal is highly effective: SOSecure consistently outperforms progressively stronger CWE-based baselines under the same retrieval framework. Importantly, the system maintains a 0\% introduced vulnerability rate and consistently improves fix rates across datasets, suggesting that any noise in the retrieved discussions does not degrade security in practice.
>
> **On Prompt Construction and Transparency**
> The retrieved Stack Overflow answer and its associated comments are appended to the prompt alongside the generated code snippet. The model is then instructed to review the code for security issues in light of the retrieved discussions and to revise it if vulnerabilities are identified while preserving functionality; otherwise, return the code unchanged. Full prompt templates are provided in our replication package, and we are happy to include representative examples in the paper to improve transparency and reproducibility.
>
> **On Evaluation Limitations**
> We agree that static analysis tools are imperfect. However, CodeQL and Bandit are standard in prior work, and all methods are evaluated under identical conditions. The improvements therefore reflect relative gains under a consistent evaluation protocol. Moreover, the gains are consistent across three datasets and two programming languages, providing additional evidence that the observed improvements are not artifacts of a particular tool or dataset.
>
> **On Functional Correctness**
> We acknowledge that systematic verification of functionality is important. To guard against regressions, we manually inspected a random subset of outputs per dataset and found that the revised code preserved intended behavior while fixing the vulnerability. We view large-scale automated functional testing as an important direction for future work.
>
> **On Model Choice**
> SOSecure is designed as a model-agnostic, inference-time safety layer. The improvements observed with GPT-4 demonstrate that even strong models benefit from external, code-specific security signals. Because SOSecure operates independently of model internals, its benefits are complementary to model scaling, and we expect similar or stronger gains when applied to more advanced models.
>
> Overall, we believe these results support SOSecure as a practical and effective inference-time approach for improving the security of LLM-generated code.

---

### Official Review · Reviewer_xHDq · 2026-03-11

**Rating:** 2
**Confidence:** 3

**Review:**

Thank you for submitting to AIWare 2026. The paper addresses an important problem. Leveraging Stack Overflow discussions as contextual signals for revising insecure code is an interesting idea, and the results suggest potential improvements. However, several aspects of the work require further clarification.

First, the motivation is not entirely clear. Why is Stack Overflow particularly suitable as the knowledge source compared to alternatives such as CWE/CVE databases, security documentation, or curated vulnerability datasets? What unique characteristics of Stack Overflow discussions enable better security guidance than existing RAG-based approaches?

Second, the novelty of the approach appears limited. The overall design resembles a standard retrieval-augmented generation pipeline where external knowledge is retrieved and used to guide LLM revision. Is the main contribution simply the choice of Stack Overflow as the retrieval source? What methodological differences distinguish this work from existing RAG-based security systems?

Third, the reliability of Stack Overflow as a knowledge source raises validity concerns. Stack Overflow content can be noisy, outdated, or context-dependent. Is keyword filtering and minimal community signals sufficient to ensure the quality of retrieved security advice? How does the system mitigate the risk of retrieving incorrect or insecure guidance?

Fourth, the evaluation relies primarily on static analysis tools such as CodeQL and Bandit. Given that these tools can produce false positives and false negatives, how reliable are the reported improvements? Does the revised code preserve the intended functionality, and how is this verified?

Finally, the efficiency and deployment cost of the approach are not thoroughly analyzed. Since SOSecure introduces retrieval steps and longer prompts, what is the impact on latency, token usage, and overall inference cost? Are the improvements worth the additional overhead?

**Summary:**

This paper proposes SOSecure, a RAG approach to improve the security of LLM-generated code by leveraging community knowledge from Stack Overflow. The system constructs a security-focused knowledge base from Stack Overflow answers and comments that mention vulnerabilities or insecure coding practices. SOSecure retrieves relevant discussions and incorporates them as contextual guidance for the model to review and revise the code. The approach is evaluated on three security benchmarks and achieves higher vulnerability fix rates compared to prompting-based baselines.

---

> ### Author Response · Authors · 2026-03-21
>
> Thank you for your thoughtful comments. We address your concerns below.
>
> **Why Stack Overflow**
> The motivation is not simply to introduce another data source, but to
> leverage a qualitatively different type of signal. CWE/CVE databases and
> security documentation primarily provide *descriptive* vulnerability
> information, whereas Stack Overflow provides code-grounded, contextual
> critiques of how vulnerabilities arise and are fixed in practice. These
> critiques are tied to concrete implementations and often include
> framework- and API-specific guidance that abstract descriptions do not
> capture. This distinction is reflected empirically: Stack Overflow
> retrieval consistently outperforms progressively stronger CWE-based
> baselines (CWE identifier $\rightarrow$ description $\rightarrow$ code
> examples), despite all methods operating under the same
> retrieval-augmented framework.
>
> **Novelty**
> SOSecure differs from standard RAG pipelines in two key ways.
> First, retrieval is conditioned on LLM-generated code and applied
> post-generation, rather than on the original prompt; this enables
> identification of vulnerabilities that emerge during generation and are
> not fully specified in the input prompt. Second, the retrieved signal
> consists of community-authored critiques tied to concrete code, rather
> than curated descriptions or labeled examples. The combination of these
> two design choices defines a distinct inference-time revision setting.
> The contribution is therefore not simply the choice of Stack Overflow as
> a corpus, but the introduction of a post-generation, code-conditioned
> retrieval paradigm grounded in practitioner-authored critiques, which
> acts as a targeted activation signal for latent model security knowledge.
>
> **Noise and Reliability**
> We agree that Stack Overflow can contain noisy or outdated content.
> SOSecure mitigates this through three complementary mechanisms.
> First, retrieval is based on code-level lexical similarity via BM25,
> ensuring that retrieved discussions share concrete API calls, library
> functions, and configuration patterns with the generated code. Second,
> we apply lightweight filtering using community upvotes to remove clearly
> unendorsed content. Third, retrieved discussions are provided as context
> rather than instructions: the model may choose not to modify the code if
> the retrieved guidance is not applicable. Critically, SOSecure maintains
> a 0\% introduced vulnerability rate across all datasets, as measured by
> CodeQL, indicating that noisy retrieval does not degrade security in
> practice. The consistent improvements across datasets and vulnerability
> types suggest that the signal from community discussions outweighs
> potential noise.
>
> **Evaluation via Static Analysis**
> We agree that static analysis tools are imperfect. However, CodeQL and
> Bandit are widely adopted in prior evaluations of LLM-generated code,
> and all methods in our study are evaluated under identical conditions.
> The reported improvements therefore reflect consistent relative gains
> under a shared evaluation protocol rather than artifacts of any
> particular tool. Notably, SOSecure achieves substantial improvements in
> fix rate without introducing new vulnerabilities, suggesting that the
> gains are not explained by false negative shifts in the static analyzer.
>
> **Functional Correctness**
> We acknowledge that systematic verification of functional correctness
> after repair is important. To guard against trivial regressions, we
> manually inspected a random sample of revised outputs per dataset and
> verified that repairs preserved the intended functionality while
> addressing the flagged vulnerability. We did not observe cases where
> SOSecure broke functionality in these checks. Incorporating automated
> testing or differential execution to provide formal guarantees is a
> natural direction for future work, and we have noted this explicitly in
> the paper's limitations section.
>
> **Efficiency and Deployment Cost**
> SOSecure introduces a lightweight post-generation step whose overhead is
> primarily determined by prompt length. On average, a single retrieved
> discussion adds approximately 530 input tokens per query, and the number
> of retrieved neighbors $k$ can be adjusted to control this cost
> directly. This overhead remains modest compared to alternatives such as
> fine-tuning or multi-sample generation strategies. Given the substantial
> improvements in fix rate (e.g., $+59.2\%$ absolute on LMSys), we
> consider this a favorable and practical trade-off for security-sensitive
> deployment settings.

---

### Official Review · Reviewer_zrT7 · 2026-03-11

**Rating:** 2
**Confidence:** 5

**Review:**

I appreciate the efforts of collecting Stack Overflow Q&As for the knowledge base construction, as well as the effectiveness of SOSecure. While I still have some concerns regarding the work.

1. The novelty of this work is limited as there are already many works building knowledge databases for vulnerability related tasks and it seems the only novelty is building the knowledge database on Stack Overflow.

2. Baselines. The authors compared SOSecure with some GPT-4 baselines, while according to the related works, there are already many retrieval-augmented generation for security, it is very questionable why these work is not compared while only compare GPT-4 baselines.

3. Dataset Selection. The authors have selected three databases which are constructed by data with explicit CWE information, it is very possible these vulnerabilities are already discussed on Stack Overflow, which makes it very straightforward for LLM to retrieve relavant information, I would suggest the authors to at least test on some vulnerabilities that are not discussed yet on Stack Overflow.

4. The authors also claimed that using Stack Overflow to construct the knowledge database is one of their core contribution, while in this case, the advantage of this knowledge database should be further explored, for instance, I would suggest the authors to evaluate SOSecure on some security bugs, which are widely discussed in the community while less concerned by existing tools.

5. The evaluation results could also be influenced if all testing databases have been seen by LLMs, I would also suggest the authors to add another dataset that are built on latest vulnerabilities or security bugs that LLMs have not seen them in their training data.

**Summary:**

This paper introduces SOSecure, a retrieval-augmented approach for improving security in LLM-generated code using community knowledge from Stack Overflow, which constructs a securit-yoriented knowledge base from posts and comments containing explicit security warnings, retrieves relevant discussions, and incorporates them as context during code revision. The evaluation across three datasets and two languages demonstrates the effectiveness of SOSecure in mitigating security vulnerabilities.

---

> ### Author Response · Authors · 2026-03-21
>
> Thank you for the insightful feedback. We address the main concerns below.
>
> **Novelty**
> While prior work explores RAG for security, SOSecure differs in a key way: retrieval is conditioned on LLM-generated code and applied post-generation rather than at prompt time. This enables identification of vulnerabilities that arise during generation and are constructed from a fundamentally different data source and retrieval setting. Critically, SOSecure leverages community-authored critiques, particularly comments tied to concrete code snippets, which provide fine-grained, framework-specific security guidance that is difficult to capture with abstract CWE descriptions. The value of this signal is validated empirically by our progressively stronger CWE baselines.
>
> **Baselines and Prior RAG Work**
> We agree that prior RAG-based systems such as Vul-RAG and VulRepair are relevant references. However, direct comparison is not straightforward. Vul-RAG is designed for vulnerability detection rather than code revision, and its CVE-linked knowledge base is constructed from a fundamentally different data source and retrieval regime. VulRepair requires fine-tuning on C/C++ data, making it incompatible with our Python-focused zero-shot setting; we nonetheless include it as a baseline on the C subset (Table 3), where SOSecure substantially outperforms it (73.3\% vs.\ 13.3\% fix rate). To provide a fair and controlled comparison of retrieval quality in our post-generation setting, we construct progressively stronger CWE-based baselines (CWE identifier -> description -> code examples), which isolate the contribution of community-sourced signals.
>
> **Dataset Overlap with Stack Overflow**
> We acknowledge that some SO discussions may reference similar vulnerability patterns to those in our benchmarks. However, our retrieval mechanism operates on code-level similarity, not dataset identifiers or CWE labels. The strongest evidence against an overlap-driven explanation is that CWE-based baselines, which have direct access to ground-truth vulnerability labels, still substantially underperform SOSecure (e.g., 87.9\% vs.\ 96.7\% on LMSys). If retrieval gains were explained by surface overlap, structured CWE retrieval would be expected to outperform SOSecure, yet it does not. The advantage of SO is most pronounced on framework-specific vulnerabilities (e.g., CWE-078 involving *shell=True* or Flask debug mode), where community comments provide actionable, code-grounded critiques rather than abstract descriptions.
>
> **Advantage of Stack Overflow Knowledge**
> The core contribution is not retrieval per se, but the nature of the retrieved signal. Stack Overflow comments provide targeted, code-linked critiques authored by practitioners in context, representing a form of grounded security reasoning that CWE documentation does not capture. Our qualitative analysis illustrates how these nudges translate directly into specific code edits, including cases where SOSecure correctly generalized a community warning about *eval* to the related risk of *pickle*, even when no exact match was retrieved.
>
> **Training Data Leakage**
> GPT-4's high No-Change Rate (NCR) across all datasets (62.5\% on LMSys without retrieval) indicates that the model frequently recognizes the code but does not act on its security knowledge without explicit inference-time guidance. This suggests that the bottleneck is not knowledge availability but *activation*: the model possesses relevant knowledge but does not reliably apply it without targeted guidance. We agree that evaluation on recently disclosed vulnerabilities with confirmed absence from LLM training data would further strengthen this argument and consider it important future work.
>
> Overall, these results suggest that the key contribution of SOSecure lies in improving the quality and grounding of retrieval signals, rather than retrieval alone.